# A Closer Look at the Quest for an Inclusive Research Project: 'I Had No Experience with Scientific Research, and then the Ball of Cooperation Started Rolling'

**Miriam Zaagsma** [1,2,*] , **Mark Koning** [1] , **Christien van Andel** [1] , **Karin Volkers** [1] , **Alice Schippers** [3,4]
and **Geert van Hove** [5]

1  Philadelphia Care Foundation, 3811 MZ Amersfoort, The Netherlands; mark.koning@philadelphia.nl (M.K.);
   andelc@philadelphia.nl (C.v.A.); k.volkers@philadelphia.nl (K.V.)
2  Department of Ethics, Law and Humanities, Amsterdam Public Health Research Institute,
   Amsterdam University Medical Center, 1081 HV Amsterdam, The Netherlands
3  Disability Studies in The Netherlands, 3453 NW De Meern, The Netherlands;
   alice.schippers@disabilitystudies.nl
4  University of Humanistic Studies, 3512 HD Utrecht, The Netherlands
5  Department of Special Needs Education, Ghent University, B-9000 Gent, Belgium; geert.vanhove@ugent.be
*  Correspondence: miriam.zaagsma@philadelphia.nl

**Abstract:** The original adage of the movement of people with disabilities 'Nothing about us without us' is fortunately more and more adopted in the research world. There is, for example, increasing recognition of the importance and value of actively involving people with intellectual disabilities in research projects on topics that are relevant to them. In a current doctoral research project, a co-researcher with an intellectual disability was recruited to work together with the doctoral researcher. Now that this project is nearing completion, it is time to look at some aspects of their collaboration and see what we can learn from this process. In several (joint) meetings, the researchers reflected on their personal experiences with working and researching together. Our reflections are presented using three overarching themes: preparations for the collaboration, collaborating as a complex process, and conducting research together. The discussion focuses on what can be inferred from these personal experiences with regard to the following three topics: how inclusive research can be organised best, the possible benefits of the collaboration for the researchers involved, and the possible impact of the collaboration on the quality of the research.

**Keywords:** inclusive research; participatory research; intellectual disabilities; collaboration



## 1. Introduction

The position of people with intellectual disabilities in scientific research has changed significantly in recent decades. Whereas for a long time they were not involved in research (others spoke for them), since the end of the 1990s, we have witnessed efforts to take a different approach. Their involvement in this regard has morphed from being exclusively research participants to gradually becoming more actively involved in the various stages of the research process. Bigby et al. (2014a) differentiate between three different approaches to how people with intellectual disabilities are actively involved in research: (1) an advisory approach (people with intellectual disabilities provide advice to academic research teams), (2) a leading and guiding approach (people with intellectual disabilities initiate, lead, and execute their own research about issues important to them), or (3) a collaborative group approach (people with intellectual disabilities work in an equal partnership with academic researchers). Each of these approaches creates opportunities, in different ways and to different degrees, for people with intellectual disabilities to influence research and help take decisions relating to aspects such as research topics, design, and used methods.

The literature on this topic cites various good reasons why it can be both important and valuable to have people with intellectual disabilities take an active role in research projects. For example, the UN Convention on the Rights of Persons with Disabilities (United Nations 2006) emphasises the right of people with disabilities to be involved in issues affecting them. In addition, being actively involved in scientific research can have a beneficial effect on the individuals directly involved (e.g., learning new skills, gaining insight into the experiences of (other) people with intellectual disabilities) and by extension on other people with intellectual disabilities and by extension on people without disabilities working with the latter (Frankena et al. 2015; Stack and McDonald 2018). It is also experienced that involving people with intellectual disabilities can contribute to the quality and relevance of research. For example, researchers with an intellectual disability can have a 'technical' contribution by developing materials that are appropriate for the research and its participants (Nind 2014; Frankena et al. 2015; Puyalto et al. 2016). Individuals with intellectual disabilities are also recognized as advocates who can help to concretize more abstract terms such as autonomy, empowerment, participation, and inclusion, bring good practices to the fore, and who take the lead in uncovering barriers (regarding, for example, accessibility, supportive relationships, or transition to adulthood) (Chalachanová et al. 2021).

When people with intellectual disabilities take on an active role, the term often referred to is inclusive research. In an article from 2018, Walmsley et al. attempted to define a 'second generation' of inclusive research, taking into account the evolutions that have occurred since the beginning of the 21st century. According to these authors, inclusive research can be described as follows (p. 758):

- Research that aims to contribute to social change, that helps to create a society, in which excluded groups belong, and which aims to improve the quality of their lives.
- Research based on issues important to a group and which draws on their experience to inform the research process and outcomes.
- Research which aims to recognize, foster, and communicate the contributions people with intellectual disabilities can make.
- Research which provides information which can be used by people with intellectual disabilities to campaign for change on behalf of others.
- Research in which those involved in it are 'standing with' those whose issues are being explored or investigated.

Practices of inclusive research are increasingly being realized in different places around the world and with different 'target groups'. In this regard, we are inspired by research centres and groups with years of experience. The fact that they have realized many projects, built a large network, presented and published a lot but also (and above all) that they have been able to exert a lot of influence on local practices and policies offers opportunities to learn from them. We would like to briefly describe two inspiring examples below. To begin with, in Ireland, we can learn from the intense cooperation between the National Federation of Voluntary Service Providers, Trinity College Dublin, and University College Cork. García Iriarte et al. (2014) and Salmon et al. (2018) report on the pathways this network developed to conduct research about topics that are important for persons with disabilities. They worked with training workshops, organized a continuous dialogue about their projects, used creative tools (such as role plays) to develop skills of researchers and co-researchers, worked hard to make sense with the teams of the data, and presented their work in local, national, and international meetings. At the same time, these researchers are able to keep a very critical stance towards their own work not wanting to become the very type of research it aims to challenge (Salmon et al. 2018). A second example is located in the USA, where Nicolaidis et al. (2019) report on the practices of AASPIRE-USA regarding trying to develop guidelines for the promotion of the inclusion of autistic adults as co-researchers. They put a strong focus on being transparent about partnership goals, clearly defining roles and choosing partners, creating processes for effective communication and power sharing, building and maintaining trust, disseminating findings, encouraging community inclusion, and fairly compensating partners. It is important to learn that for persons with autism (the

group that is often strongly entwined in a clinical model), the time has come to participate in research. The lessons learned in research projects with people with autism can help us to make the framework for research projects and the communication about the projects clearer for colleagues-researchers with intellectual disabilities as well.

As a result of the growing attention to inclusive research and the increase in research initiatives with an inclusive approach, the knowledge shared (through publications) on this topic is growing. The publications can be roughly divided into two groups. There are articles that primarily focus on personal experiences of inclusive research and reflections on collaborative research (e.g., Strnadová et al. 2014; Dorozenko et al. 2016; Riches et al. 2017). In addition, other publications attempt to arrive at assertions that can be generalised across inclusive research initiatives, such as attributes that should be taken into consideration when conducting inclusive research (Frankena et al. 2018), competencies that are considered important for researchers with and without intellectual disabilities in inclusive research initiatives (Embregts et al. 2018), and contextual and team-level factors and processes that foster and maintain inclusive research (Schwartz et al. 2020).

Through this article, we want to align ourselves with the tradition of incorporating personal experiences. For the first two authors (both researchers, of which one has an intellectual disability), this article represents the culmination of an extended period of collaboration within a research project (see Section 2.2). The first author positions herself in a tradition of doctoral students opting for a more inclusive approach in their research work. Already in 2008, Björnsdóttir reported on the tensions that inclusive research during a PhD trajectory evokes within a competitive academic environment and the danger of academic researchers falling into the same trap of the exclusion that they criticize society for (Björnsdóttir and Svensdóttir 2008). By focusing on the tension between inclusive research and traditional ethical guidelines at research institutes and universities, Morgan wanted to make future PhD researchers aware of possible incompatibilities linked to, e.g., disclosure, the tension between empowerment and protection, and the application of shared partnership, equality, and transparency within an academic context (Morgan et al. 2014). Moreover, Dorozenko's recent call for PhD students working with inclusive research to have sufficient reflexivity and critical reflection (e.g., regarding the risks of repeating oppressive power relations) was very inspiring for the collaboration described here (Dorozenko et al. 2016).

The aim of this article is to reflect on our own research collaboration by exploring our personal experiences of how our collaboration in research works best, as well as the benefits that our collaboration has brought to us personally as well as to the quality of the research. We hope that our experiences can provide support to other researchers who wish to set up and conduct inclusive research. This article was realised thanks to the substantial contributions of several people who are all co-authors. Mark and Miriam form the research duo that worked together on a research project for several years (see Section 2), and for this article, they reflected on their collaboration. Geert, Karin, and Alice were involved in this research project as supervisors and advisors. Not only did they advise the research duo on 'technical' research issues but also on conducting research inclusively. Christine was asked to help the research duo compile and situate experiences (see Section 3). With regard to the writing of this article, Geert, Miriam, and Mark took the lead, and the other three co-authors reviewed previous versions of this article.

## 2. Context

### 2.1. The Research Duo

Mark and Miriam both work as researchers at the Philadelphia Care Foundation (PCF) in the Netherlands. The PCF is a care organisation for people with an intellectual disability, which offers a wide range of support services throughout the Netherlands. Over a period of almost four years (December 2017–October 2021), Mark and Miriam worked together on a research project into the experiences with an online support service of people with intellectual disabilities living independently (more on this in the next section). In addition,

Mark works several hours a week as an assistant on other projects within the PCF. Mark and Miriam introduce themselves below. Mark: *'I am 45 years old. I was born prematurely and for a long time I believed I had a developmental disorder. I went to a special education school. For a long time, I felt like I had a hard time making friends, but I did have contacts with other people. Once I finished school, I had various jobs in administration. I often felt like I wasn't taken seriously at work. I also had the feeling that I was only half-participating in society. I had the feeling that something was wrong, but I did not know what it was. In 2005 I was diagnosed with PDD-NOS, a form of autism. That's how I ended up coming into contact with Philadelphia* [the service organisation]*. Then things started to change. There were opportunities at work to focus on my talents in a partially sheltered way, I was able to develop myself. This was enhanced when I joined various client councils within the organisation. I felt I was being taken more seriously. I mattered. Through people at the client council, I came into contact with Miriam in 2017. She was looking for a colleague to conduct research together. At the time, I had no experience with research, let alone scientific research. Back then, I couldn't have told you what research involved. And then the ball of cooperation started rolling'.* Miriam: *'I'm 40 years old. I went to a mainstream school and completed studies at university, where I was able to gain a lot of knowledge and skills in the area of scientific research. After my studies, I did various jobs conducting research. Sometimes these were projects in which we tried to involve the 'target group', such as young people in mainstream education. However, these were not truly inclusive projects. As such, when I joined Philadelphia in 2016 I had no experience with inclusive research'.*

### 2.2. The Research Project

The project that Mark and Miriam worked on involved research into the online support service DigiContact. This service is offered by the PCF as part of a broader package of support services for people with intellectual disabilities living independently (in their own homes) (Vijfhuizen and Volkers 2016). DigiContact offers 24/7 remote support, where people with a support need can contact a team of specially trained support workers via either an app or link on their mobile phone, tablet, or computer or via a standard telephone. The aim of the project was to compile knowledge on the experiences of both support users and professionals of DigiContact regarding the potential value of its support for people with intellectual disabilities living independently. During the course of the project, five sub-studies were performed of which each focused on a different question. A scientific article was (or will be) drafted in English on each sub-study (Zaagsma et al. 2019, 2020a, 2020b, 2021).

The research project started in 2015 with a different researcher duo. After one year, both members of this duo left the project because neither of them wanted to continue working as researchers. After this, Miriam started working on the project in February 2016. During the first year, she worked with another co-researcher for several months. Mark was recruited as a co-researcher in 2017. For Miriam, the research project formed the basis for her PhD at the Amsterdam UMC (Vrije Universiteit). Throughout the project, she was able to devote 32 h per week to the research project. When Mark was recruited, he was offered a contract to work on the research project for an average of 8 h per week. He also worked a varying number of hours on other projects within the PCF. Besides Miriam and Mark, three senior researchers (fourth, fifth, and sixth author) were involved in the different sub-studies into DigiContact. They provided advice for the research and supported Miriam in her PhD.

## 3. Materials and Methods

In this article, we reflect on the personal experiences of Miriam and Mark with conducting research together during the research project on the potential value of 24/7 online support. By doing so, we align ourselves with the tradition of an auto-ethnographic approach, in which personal experiences are described and analysed in order to understand broader cultural experiences (Ellis et al. 2010).

During the course of the research project, data on the collaboration were logged in various ways. To begin with, both Mark and Miriam independently kept a logbook in which

they wrote down their personal experiences, thoughts, questions, doubts, and difficulties on a weekly basis. During the first weeks of the project, Mark was supported by Miriam on how to keep a logbook (e.g., by discussing together what to note, when to do this, with how much detail, etc.); after this, he continued to do this by himself. In addition, Mark and Miriam had regular conversations together about the research and their collaboration in particular. These conversations were not always planned in advance and did not have any fixed structure. Notes of these conversations were recorded in the logbooks.

In preparation for this article, five meetings were organised and held (in the spring and summer of 2020), in which Miriam and Mark reflected on their collaboration under the guidance of a moderator (3rd author). This moderator was able to reflect on their experiences and ask questions from an outsider's perspective. At the start of these meetings, the moderator, Miriam, and Mark discussed topics that would be interesting to explore. Examples of topics were: expectations of conducting research together, how different research activities (e.g., analysing data) were carried out, and perceived facilitators and barriers in the process of collaborating. Before each meeting, the moderator prepared a list of questions she could use to fuel the conversation when needed (e.g., 'What were difficult moments during the project (and why)? What are you proud of (and why)?'). The notes in the logbooks were used as input for the meetings, with Mark and Miriam going through them in advance to refresh their memories. To prepare for this, a plan of action was drawn up by the researchers together. This plan included a guideline regarding what to do (e.g., what to pay attention to, how to make notes of topics that seemed important) and a plan on when to do this (the work was spread out over multiple shorter sittings because the collection of notes was quite comprehensive). Following up on this plan, the researchers worked independently from each other to review their own notes without needing further support. The five meetings took place remotely via videoconference, on account of the COVID-19 measures in place at the time. Three of the five meetings were conducted jointly. In the other two meetings, the moderator talked with each researcher individually, focusing in particular on elements that were related to Mark and Miriam's unique and highly personal perspective. Each meeting was recorded and subsequently transcribed by an external agency.

The transcripts of these five meetings and the logbook notes formed the research material. To decide on which reflections and experiences to present, Miriam and Mark went through the material while keeping the three topics of reflection in mind (i.e., how inclusive research can be organised, the possible benefits of the collaboration for the researchers involved, and the possible impact of the collaboration on the quality of the research). They started this process separately from each other. Both of them read all the transcripts and notes. To assist with ease of comprehension, Mark also listened to the audio recordings of the meetings, as this helped him retain his attention and grasp the meaning of what was being said. They both used marker pens to highlight parts of text on a given topic, always writing the gist of the text in the margin. They each made a list of the experiences they personally found important. These lists were shared with each other by e-mail. As a next step, Miriam and Mark compared and discussed the experiences on their lists in order to come up with a joint overview in which their experiences were integrated. This process took place (mostly) via videoconference calls due to COVID-19 restrictions being still in place. This overview was shared with the other authors in order to decide together with them on which experiences needed to be highlighted in this article. The starting point in this was to present experiences that we were convinced would interest other individuals involved in inclusive research.

## 4. Results

The experiences are discussed and explained on the basis of excerpts from the meetings. These excerpts were translated into English by a certified translator. The experiences are presented using three overarching methodological themes: preparing for the research collaboration, collaborating as a complex process, and conducting research together.

### 4.1. Preparing for the Research Collaboration

The first methodological theme describes the period leading up to the collaboration and emphasises the importance of good preparation.

#### 4.1.1. Philadelphia Opts for Inclusive Research

The decision to adopt an inclusive approach in the research project did not appear to be a priority for either researcher in the first instance. The decision to set up and conduct inclusive research was not made by the researchers themselves. It was the steering committee of the research project (with representatives from the PCF and the senior researchers from the university) that put forward and decided on the idea of inclusive research at the outset of the project. Both Miriam and Mark were candid and stated that at the time of their application, they were primarily interested in the topic of the project and the accompanying work activities. The inclusive nature of the research project was for them an attractive and interesting extra. Miriam: *'The position first attracted me because of the subject of the research project: the digitisation of care for people with intellectual disabilities, and studying client experiences. The possibility of obtaining a PhD on the subject was also interesting . . . I didn't have any experience with inclusive research, although I did have ideas and expectations about it. I saw it as a nice challenge to take on, and an enriching experience. I expected to learn from my colleague what it is like to live and work with an intellectual disability. And that it would provide openings to make the research more accessible and allow people to participate in it'*. Mark was interested in DigiContact and wondered whether this form of support would suit him if he lived independently. Above all, Mark was looking for ' . . . *a job that would give me more influence. The job was also all the more interesting because it was a paid job.* [Mark was already doing certain tasks for the PCF, but these had been voluntary up until that point]. *Plus, I am the kind of person who is always eager to learn, and I am interested in other people. The research project gave me the opportunity to meet other people. Working together intensively with someone on a new area of work also gave me a safe feeling'*.

#### 4.1.2. The Search for a Good Research Duo

Making the decision to set up and conduct inclusive research is one thing. We learned in this research project that everything stands or falls with the quality and sustained commitment of the research team. The research project we report here was suspended on two occasions after people quit the project. The first research duo (the predecessors of Miriam and Mark) quit at the same time because neither of them wanted to continue working as researchers. After Miriam started on the project, she worked briefly with another co-researcher before starting work together with Mark (and finishing it together). This collaboration ended after several months because the role of co-researcher did not match well with their talents and ambitions, and the level of support that was needed to enable their participation in research activities was beyond what the organisation and Miriam could provide. During this collaboration, there was no job coach involved, as was the case in the collaboration between Mark and Miriam (see also Sections 4.2.1 and 4.2.4).

This difficult start meant that in the search for replacements, other strategies were tried out in order to put together a research duo that could collaborate effectively together. For example, the initial approach for recruiting the co-researcher was to disseminate flyers with accessible information about the research project and the role of the co-researcher. The candidates who applied were interviewed and hired by staff from the PCF but not by the other researcher (Miriam). When the search for a co-researcher started again, a more focused approach was taken, and Miriam was involved in the interviews with candidates. The support person for the client councils at the PCF (someone with a good idea of the ambitions and talents of clients) had focused interviews with persons whom he thought would not only be suitable but would also be interested in the research project. With regard to suitability, specific attention was paid to verbal and social skills (with a view to conducting interviews) and the ability to travel independently (e.g., by public transport). Mark was seen as a highly suitable and motivated candidate. Following both a telephone

conversation and face-to-face contact between Mark and Miriam, they concluded that their expectations matched well. Moreover, both had a good feeling about these first contacts and could see themselves working together intensively for a longer period of time. '*It was immediately obvious to me that it was a good match*', says Mark on the subject.

The first time they clicked turned out to be an important basis for the subsequent collaboration. According to Miriam: 'When we needed to search for something, or when things weren't going so well, we could always fall back on the experience of that first connection. For example, issues were discussed openly and we were able to look for solutions together in an open and pleasant way'.

### 4.2. Collaborating as a Complex Process

An organisation may decide to organise and facilitate inclusive research, and researchers may decide to participate in it, but this is no guarantee that the research will run smoothly. The second methodological theme addresses the difficulties we experienced in working together as a research duo.

### 4.2.1. Time (and a Lot of Organising) as an Important Factor

As happens with many employees, Mark had to make arrangements with his former employer before he could join the research team. The fact that Mark, as a person with a disability, could not simply decide to change jobs is notable in this regard. He explains: '*After the job interview for the position of co-researcher, and the happy moment when I heard that I had been hired, I immediately started to find out whether it was administratively possible to take on the job . . . As someone with a disability, I have to deal with the UWV[1]. This is a good institution because it ensures that I can participate in the labour market. But at that point it was also holding me back from self-development. Even though I felt I no longer fitted in at my previous job, the UWV felt that my permanent job was more important because it offered me income security. Fortunately, Philadelphia identified this problem and enlisted the help of a job coach. Despite the mediation of the job coach, we could not get in contact with my then employer. In the end, Philadelphia started looking for additional projects for me within the organisation alongside the role of co-researcher; that way the job coach had a credible story and convinced the UWV that this new workplace was suitable for me. This all took four months. I remember these months as a period with a lot of uncertainty, disappointment and hope. In the meantime, Miriam and I were in touch every now and then. We also went out to dinner once. When we met up, we discussed everything that had to be arranged, but Miriam also talked about the state of play of the research. As such, we were both up-to-date with the situation*'. All this shows that finding people with disabilities to perform the role of co-researcher can be difficult, even if it is a paid job. The system gives priority to security of income rather than self-development and learning new skills. It was a tense period for Miriam as well, as conducting inclusive research with the co-researcher not being able to actively participate yet was not easy. Fortunately, her manager at the PCF took care of all the 'organising work' so that she could focus on the research work.

As we can see, this research project had a challenging start. Miriam was obliged to keep waiting for her co-researcher and had to manage the first sub-study and the data collection for the second sub-study on her own. Mark was on the sidelines and witnessed a few developments (including a series of interviews) that he would have preferred to have been involved in. Despite the fact that Mark was given a lot of time to familiarise himself with the subject of the research at the start, it is clear that analysing research data that one has not collected themselves is not the best and most pleasant way to join a project. Nevertheless, Mark says that he has good memories of his first weeks in the research project. He felt he could make a contribution for other people with disabilities and that the research was meaningful and not just for himself. Mark felt highly appreciated and was impressed by the fact that so many people were interested in the research (he participated in a working group and attended a conference): a new world appeared to be opening up.

### 4.2.2. Getting to Know Each Other Really Well

The first weeks in which Miriam and Mark worked together were crucial in laying the groundwork for a good collaboration. They took the time to get to know each other well. Mark was introduced to the rest of the department. Moreover, his first day at work was rather special. Mark explains: *'As we had had regular contact in the preliminary stage, the first working day was a very pleasant start to the whole experience. Moreover, the first day—a Saturday—was a conference day. This took place at a hotel in Amsterdam. Miriam had to give a presentation about DigiContact. I only had to go along and listen. But once there I was introduced to everyone as the co-researcher, and in a room full of researchers! Important contacts were made that day for the research project, and I was given a lot of information, but in bite-sized chunks. It was a special start for me with all the contacts and information. That week I was also given an article in English that Miriam wrote, to read'.*

Besides this very intensive start (for Mark), Miriam and Mark learned that there is more to working together successfully. Miriam: *'It was essential to take our time, have regular lunches in addition to the necessary functional talks, or catch up on things going on in our lives outside of work. These conversations were crucial for building a good working relationship. We got to know more about each other, we learned how we could work together, and each other's habits, for example the fact that I like to have a quiet start to work in the morning, became clearer. That way, you can get on the same wavelength, and it is important to maintain a good working atmosphere'.* (Figure 1) Finding enough time to connect with each other outside of working on research activities together was sometimes challenging, for example, due to Miriam wanting to make pace in order to meet certain deadlines (related to her PhD planning) and Mark experiencing the pressure of working on several projects (besides this research project) at the same time.

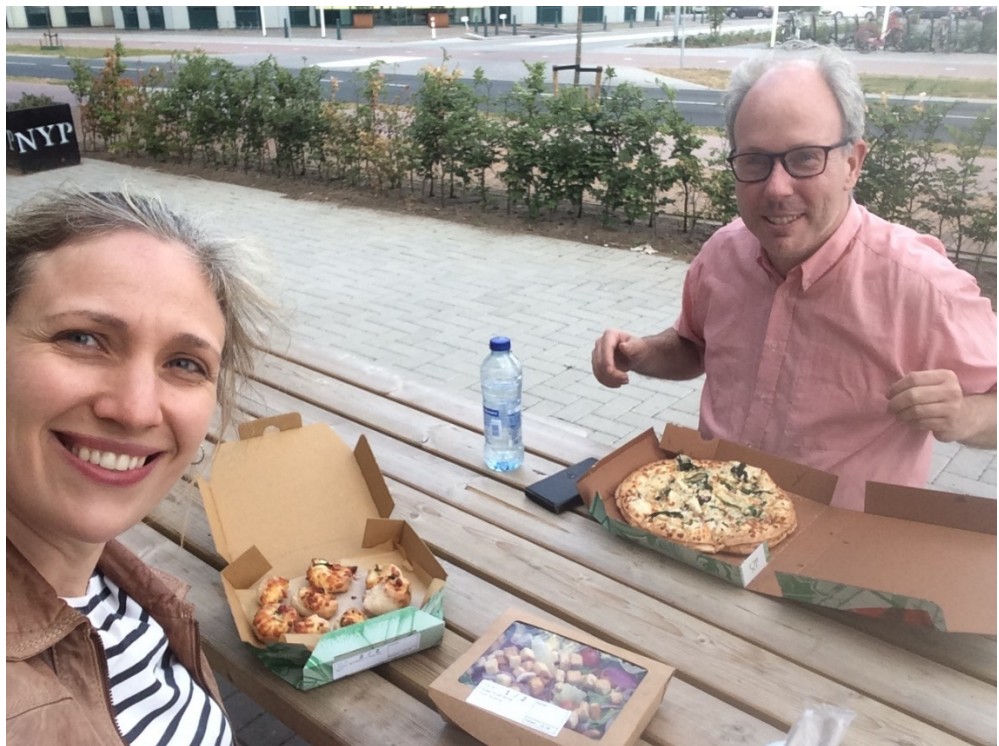

**Figure 1.** Going for pizza (10 July 2018).

Conducting research for and within a large organisation such as Philadelphia also means that getting to know that organisation and being introduced to it and its broader research network are important. Mark explains: *'Fortunately, I already knew a number of people in Philadelphia through my work in the client council. I attended a lot of meetings on the research project, and got to know many people from Miriam's research network. After a few months,*

*I was also able to make a contribution in presentations about the research project. In the beginning, I mostly talked about my role as co-researcher; but after a while I could also give more and more information about the research itself'* (Figure 2).

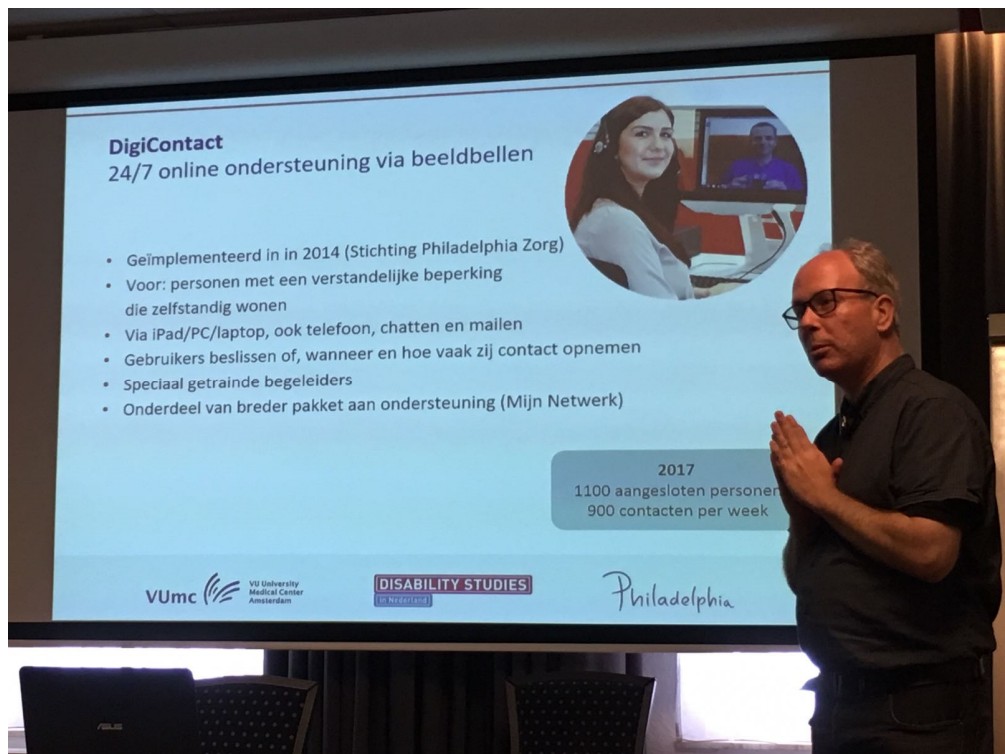

**Figure 2.** Mark presenting (21 June 2018).

4.2.3. The Need to Keep Reflecting

As Mark recounted: 'We did a lot of research together in the first few weeks. That seems normal, because I had no experience with scientific research. But the fact was, I only worked on the research project for 8 h a week, while Miriam worked for 32 h a week. So we took the time to find out where I could be most useful. It was important in this regard to look not only at what needed to be done, but also at what our interests and talents were. That's why it is so important to get to know each other. We decided not only to touch base each week regarding the practical aspects of the research project, but also to set aside time two or three times a year to discuss our collaboration in more detail. Conducting the research together was an important part of the project; so this was taken seriously. We also kept a logbook, this was also important for the research'.

In this research project, working together to a clear structure proved valuable. For example, Mark and Miriam worked together on a fixed day of the week as much as possible, and this day started with a meeting where there was an opportunity to catch up in general and to discuss the work. They always made clear agreements about who would take on which tasks. These agreements were put into an overview that was accessible to both researchers.

The flexibility that is sometimes required when conducting research was an additional challenge for Mark, as he was also working on other PCF projects. For example, the interviews could not always be planned on the fixed working day, and there were not always tasks for the full eight hours, or, conversely, there were too many tasks. Mark took up this challenge together with a job coach: this coach taught him how to make overviews of the activities he needed and wanted to do, plans were revised, and in his digital agenda, he learned to work with time blocks in different colours (each project a different colour).

### 4.2.4. Roles, Allocation of Tasks, and Decision Making

Looking back on the collaboration, a number of things stand out with regard to roles, allocation of tasks, and decision making. For example, before starting the collaboration with Mark, Miriam had had a difficult experience working with a previous co-researcher, which stayed at the back of her mind and made her feel unsure about working with a co-researcher. It is clear that the PCF played a highly valuable role in this regard: the foundation created the conditions that facilitated both parties in their collaboration. For example, by hiring a job coach for Mark, Miriam could be a colleague for him and did not (also) need to take on the role of a support worker. By finding several projects for Mark, he could get started without being thrown in at the deep end.

Mark and Miriam allocated tasks in mutual consultation. As Mark had substantially less time available for the project than Miriam (8 versus 32 h per week), it was not possible for him to be involved in all activities in the same way. In this respect, a distinction was made between activities that they performed either together or independently from each other and activities in which Mark adopted the role of advisor. The starting point for this distinction was their personal skills and interests, as well as the expectations regarding in which activities Mark's participation and input would add the most value to the project. Miriam explains: '*Mark discovered he really liked to work on data analysis, and we both felt that his involvement led to broader and richer insights. This made us decide that Mark would spend a relatively large part of his time on analysing, and less on, for example, writing texts. Making such decisions was sometimes difficult, as in some situations Mark wanted to be involved in something, I remember specifically one time when we had to prepare for a presentation, but we decided that he would not be involved because there wasn't enough time.*' When several sub-studies were performed at the same time, it was easier for Mark to first finish his work on one sub-study before moving on to the next.

During the research project, Mark never took 'the lead' over a sub-study. Questions such as whether this would have produced more results, or given more scope for experimenting with different research methods, therefore remain unanswered.

### 4.2.5. Collaboration during the COVID-19 Pandemic

This research project was hit full-on by the COVID-19 pandemic. Mark and Miriam were about to start with a new round of data analysis, and they had to find a way to continue this activity remotely. They describe below how working remotely disrupted their working rhythm, appointments, and rituals. Mark: '*During the final phase of our research, the Netherlands was struck by corona. This meant that we had to work from home; I felt very limited in the options available. We drifted apart a little bit because we has less contact with each other, and we weren't able to motivate each other as much. We did call every week and we were in contact* via *WhatsApp, but we sometimes lost the focus of our research*'. Miriam: '*In the first chaotic phase of corona, we lost sight of each other for a while. We learned to find a new digital rhythm. We were lucky that many of the interviews were already completed and we already had the research material. We called each other once a week to clarify and divide up the work. It was also an opportunity to catch up. The analyses had to be carried out remotely. Fortunately, the lockdown didn't stay very strict for too long, and we were eventually able to sit together again in person. But after a few months, we had to switch back to digital as the measures were tightened again. We went back to videoconferences, and trying to find each other with regular phone times. And in the meantime, we had to continue working on our own tasks*'. Having to work together remotely guided the researchers towards analysing data more independently from each other. Before COVID-19, data analysis was largely performed together, in the same room, using post-it papers or other materials to record codes and cluster them (see also Section 4.3.2). Now the researchers had to change tactics. They read and coded the transcripts separately and independently (and without further support) from each other. Mark and Miriam were both comfortable with this, as they felt they had built up sufficient experience with coding in previous studies. They printed transcripts, highlighted pieces of texts, and wrote their codes in the margins. The clustering of codes in sub-themes and themes proved to be more

difficult to do together remotely, as they missed the (visual) grouping processes they had used before. Fortunately, this process could be picked up again after a few months when it became possible to (occasionally) work together in the same room.

### 4.3. Conducting Research Together

Conducting research involves certain activities. This third methodological theme addresses our experiences of conducting research tasks together. Two specific tasks that took an important position in several phases of the research project were chosen: interviewing and analysing.

### 4.3.1. Interviewing

For the research project, two rounds of interviews were completed during the collaboration between Miriam and Mark. In Mark's own words: '*I was going to do qualitative research, this is a term I didn't really grasp - especially at the beginning. Here, conducting research together meant first preparing questions together. These questions had to fit the research topics we wanted to know a lot about. We learned to first draw up a research question together and then turn it into research, so that we got answers to these questions. I always tried to prepare our meetings for this. I wrote down ideas to ask questions about between our meetings, so I was prepared for the next work meeting with Miriam. That way, my share in the topic list became increasingly clear. Besides consultation sessions, we also shared files within Philadelphia; we shared and emailed with each other regularly and came up with the best possible list of questions. We also drew up a letter of invitations together; I also called the potential participants to arrange appointments. At the start, I had no personal experience with interviewing, so we went to do a trial interview with someone I knew well. This data was not used for the research project. We interviewed both supervisors and clients who used DigiContact. We usually had several interviews with these clients. In between the interviews, we were able to sit together and discuss. This allowed me to grow in my role as interviewer. In the beginning, Miriam asked most of the questions and I supplemented her from time to time, but by the end the roles were reversed.*'

### 4.3.2. Analysing

As regards the analysis of the interview data, Miriam and Mark worked closely together to find the most convenient way to make everyone's share as rich as possible. Mark became a highly active researcher during the analysis (Figures 3 and 4). As he says himself: '*The analysing often started during the interview itself. We noted our observations individually during the interview, and used them as the first step in the analysis. There was an external person who would type up all the interviews for us verbatim. If these typed texts came back, they had to be anonymised; that was quite a job, a task we could divide up. Bullet-point summaries of the interviews were also made, Miriam usually did this. The people we interviewed were subsequently approached to check whether we had correctly understood them; this is referred to as a member check. Miriam went back to the professionals and I handled the clients. In-depth analysis is something that takes a lot of time. I learned that the different perspectives sometimes contrasted with each other, and I knew that I could make a difference with my perspective as an expert by experience compared to the insights of the professional researcher. We also focused a lot on the different research questions, which required going through the research material again several times. Sometimes I preferred listening to audio recordings of the interviews rather than reading the transcripts. So that's what I did. We used different methods to perform the analysis. Sometimes we wrote our findings on post-its. We also sometimes highlighted sections of interviews when we were taking excerpts from them. Whenever I learned new things in other projects, I would present them to Miriam and we would see if they could be used in our collaborative research. For example, visual analysis methods were also used: things were then grouped and pasted together like a collage. At other times, we would perform the analysis with several people on the research team and look at the data from the broader research team.*'

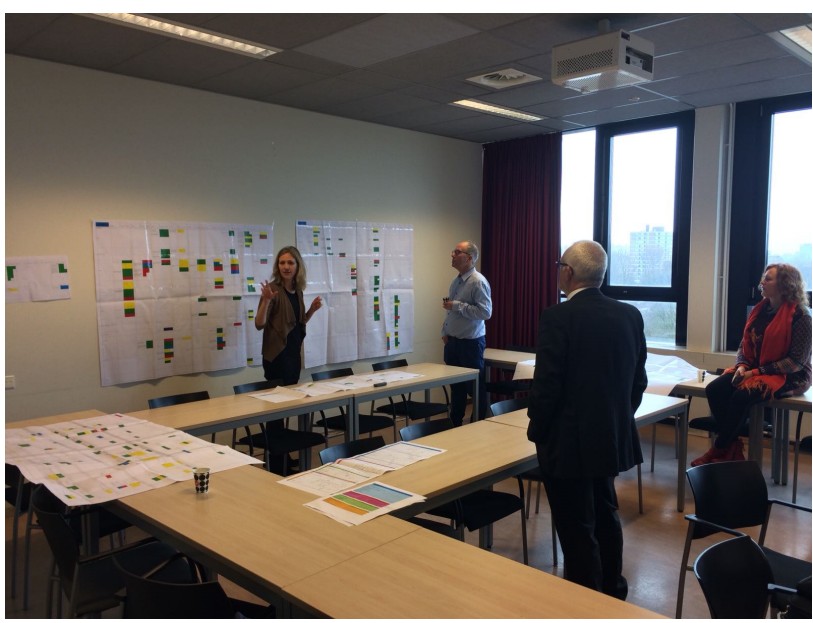

**Figure 3.** An analysis session with the entire research team (8 March 2018).

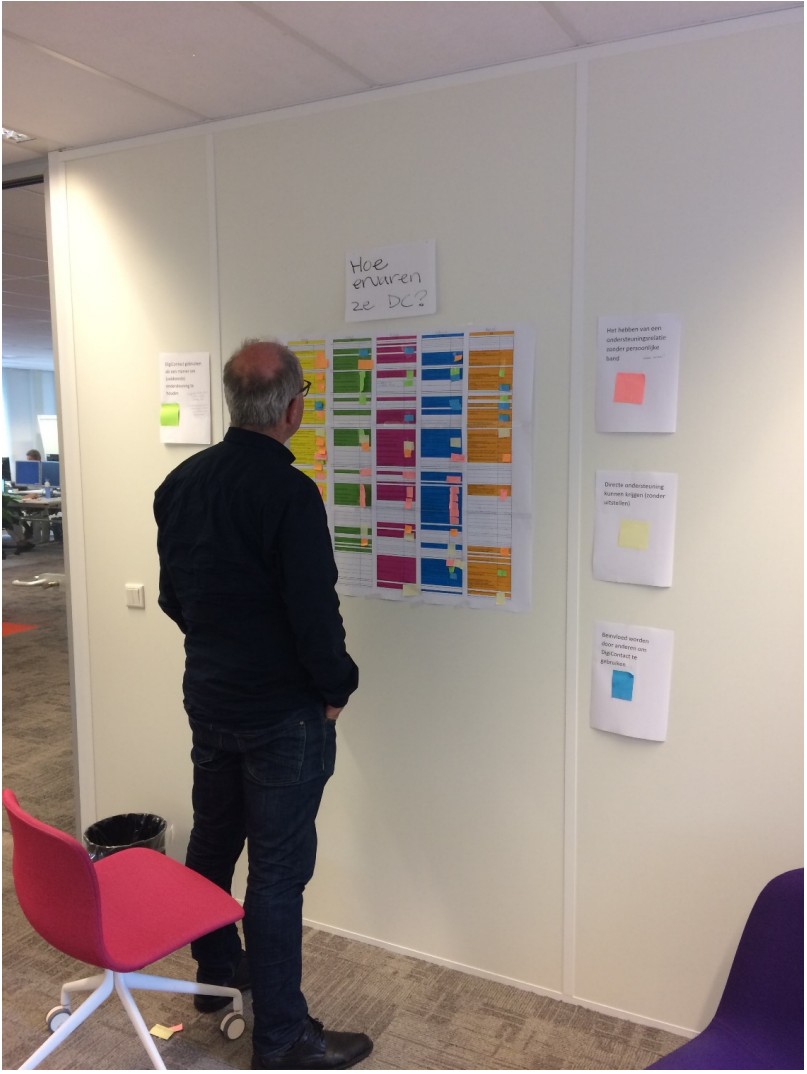

**Figure 4.** An analysis session of Mark and Miriam (3 June 2019).

**5. Discussion**

At the start of this article, we explained that we would reflect on three topics, based on our personal experiences: (1) how best to organise inclusive research, (2) the possible benefits of the collaboration for the researchers involved (Mark and Miriam), and (3) the possible impact of the collaboration on the quality of the research. In this discussion, we first look at what we can infer about these topics from the research material presented. We then briefly compare our results with the recent research literature on inclusive research.

Regarding our experiences on how to organise inclusive research, it is clear that this was difficult during the start of this research project. For example, the first research duo quit after a year, the co-researcher of the second duo quit after a few months, and after that, it took a long time before Mark and Miriam could really start working together. Our experiences led to several insights. First, we experienced that not everyone is motivated, ready, and able to become a member of a team doing inclusive research. It is therefore important to recruit researchers and co-researchers who are suitable partners in inclusive research. It also underlines the importance for organisations and research institutes that want to engage in inclusive research to provide education and training on inclusive research methodologies (García Iriarte et al. 2021) for researchers (both academic researchers and co-researchers). Second, we learned that hiring people with disabilities as paid researchers can come up against various administrative hurdles. A research job is often temporary and not always stable over longer periods of time. Many people with disabilities are caught in the 'golden safety net' of social security: they have a good monthly income; however, this can be at the expense of further self-development and taking on new challenges. Third, we learned that the party who commissions the research (in this project, a large service organisation) can play an essential role in inclusive research, a role that does not have to result in interference with the body of the research. In this project, the PCF really 'stuck its neck out' for the final duo. For example, Mark was offered more than one project to provide him with a stable income. In addition, the unavoidably slow pace of inclusive research was well understood: more time was allocated to the research project. In this project, introducing a job coach proved to be a golden asset. This job coach could work in a supportive manner in the function of what Morgan et al. (2014) named transparency. The job coach could also, especially during the first phases of the research, help to inform and clarify certain aspects. This intervention allowed both researchers to really work as colleagues, and thus this aspect of power imbalance could be mitigated.

As regards the benefits to both researchers of conducting research together, the research data show benefits to the co-researcher: Mark got a job that allowed him to have a more direct influence on the quality of life of people with disabilities. A (research) world opened up to him in which he gained respect, and he states that this had a great influence on his self-confidence and his ability to acquire new skills. He learned to listen carefully to people, ask the right questions, and see that the perspective of the experienced expert could really contribute something to the research. The academic researcher also personally benefitted from the opportunities that were created for Mark to have a voice, to exert control, and to make decisions, as this made sure that this project was not 'just another' PhD that only reflected the perspective of non-disabled experts or researchers (Björnsdóttir and Svensdóttir 2008; Dorozenko et al. 2016). The academic researcher also learned from their long-term collaboration about ways in which they could best work together and conduct research together. In this regard, it was especially difficult situations from which she could learn, such as a previous collaboration with a different researcher with intellectual disabilities in which the roles of being a supporter and being a colleague became too intertwined.

Finally, as regards the quality of the research, we would like to emphasise in particular the fact that Mark, as a co-researcher, seized opportunities through his involvement in various projects to take methods from one research project to the next. For example, methods for visually grouping data were contributed in a very creative way and provided additional depth. In this regard, we go further than Björnsdóttir and Svensdóttir (2008) who

state that well-executed inclusive research is of the same value as non-inclusive research. In our research project, the inclusive approach led to a higher value—something that can be deduced, for example, from the very high number of articles that were eventually published within this PhD. In addition, the slow pace at which the research advanced also ensured that all research steps were prepared and experienced more intensively by both researchers (see, for example, the way Mark describes how research questions and interview questions were meticulously coordinated (Section 4.3.1).

In this final section, we compare our research findings with some other sources on inclusive research. Several authors, including Nind (2011), Dorozenko et al. (2016), and Tilley et al. (2021), call for continued attention to and a critical reflection on power processes in inclusive research. In our project, Miriam was determined (after a previous rather unsuccessful collaboration with another co-researcher) not to take on a 'carer role' vis-à-vis her new colleague. Mark had to be a genuine colleague. Hiring a job coach for Mark clearly made a difference in this regard, as already discussed in the second paragraph of this discussion. At the same time, the findings also show that when a co-researcher has significantly less time available for the research project than the academic researcher, this has the potential to contribute to an imbalance in power. Besides this search for collegial roles, it is also noteworthy that Mark was able to learn a great deal during the various sub-studies regarding conducting interviews and analysing data. This contributed to feeling more self-confident about his research skills. Whereas in some projects, co-researchers are only or primarily involved in data collection, in this research project, the co-researcher was involved in all phases of research, as well as data analysis. Mark joined the project at a point when the data of a sub-study needed to be analysed. From the outset, he clearly contributed an experience-based perspective, but this came into a sharper focus when he was able to try out a number of methods (e.g., highlighting parts of the texts he was given). Still further, he was able to collaborate with several people from the research team for analysis (at these moments, the 1-on-1 with Miriam was opened up) and contributed his own analysis methods (a visual grouping method). It is clear that advancing insight, time, and growing research competencies all contributed to an increasingly equal power balance between the researchers. This brings us to the three approaches to inclusive research as outlined by Bigby et al. (2014a) and presented in our introduction. We believe we can say that, despite all the barriers, this research can for the most part be situated in the collaborative group approach (i.e., people with intellectual disabilities work in an equal partnership with academic researchers) (Bigby et al. 2014b). Mark was given the space to realise his dream of having a job that would allow him to influence the quality of life of people with a disability. He took full advantage of the opportunity to bring the perspective of the experienced expert into the research project and to influence the progress of the research. In our opinion, there were significant attempts in this research project to take seriously the (power) balance between researchers with and without disabilities.

As a final topic of the discussion of our findings, we return to Walmsley et al.'s (2018) view of inclusive research. In studying the 'second generation' of inclusive research projects, they advocated for a special consideration for the difference that can be made by co-researchers with disabilities and for the possible impact they (also) have on the quality of life of people with disabilities. In our research project, Mark joined the team with the clear intention of making a difference for people with disabilities. We believe that, partly due to the way in which he took responsibility within the project, the DigiContact support service (being the object of the research project) was put in the picture both from the perspective of experience experts and as a realistic support option. Regarding the latter, it was shown that although DigiContact is not a miracle cure that can replace all onsite support and works equally well for everyone, it does offer people an additional support alternative that can contribute to their possibilities to participate healthily in society.

When looking back at our reflection process, we feel it would have been valuable to include more people who were also involved in or had an impact on our collaboration. For us, it was an important insight that 'others' played an important role in our collaboration.

This was especially the case for the job coach who supported the co-researcher regarding various work-related issues and the organisation that commissioned the research project and created the conditions under which our collaboration was shaped. For this reason, it would have been interesting to involve these parties and also include their experiences and points of view. For future reflections on inclusive research processes and experiences, it is advisable to think (in advance) about which actors play a role in (or have an impact on) the collaboration and to include their voice in reflection processes as well.

**Author Contributions:** Conceptualization, M.Z., M.K. and G.v.H.; Formal analysis, M.Z. and M.K.; Investigation, M.Z., M.K. and C.v.A.; Project administration, M.Z.; Supervision, K.V., A.S. and G.v.H.; Validation, K.V., A.S. and G.v.H.; Writing—original draft, M.Z., M.K. and G.v.H.; Writing—review and editing, M.Z., M.K., C.v.A., K.V., A.S. and G.v.H. All authors have read and agreed to the published version of the manuscript.

**Funding:** This research received no external funding.

**Institutional Review Board Statement:** The Medical Ethics Review Committee of VU University Medical Center (FWA00017598) confirmed that official approval was not required as the Dutch Medical Research Involving Human Subjects Act (WMO) did not apply to this study. The researchers followed the Disability Studies in The Netherlands Code of Practice in Research (Disability Studies in The Netherlands 2017).

**Informed Consent Statement:** Informed consent was obtained from Mark and Miriam. Audio-recordings were made after all involved parties gave their approval and the audio files were destroyed after analysis had been completed. All data were anonymized and subsequently handled and stored with care and respect for privacy. The persons depicted in the included photo's all gave their permission to the publication of these photo's as part of this article.

**Data Availability Statement:** The data presented in this study (transcripts) are available on request from the corresponding author. The data are not publicly available due to privacy or ethical restrictions.

**Conflicts of Interest:** The authors declare no conflict of interest. The authors also declare that the funder had no role in study design, the collection, analysis and interpretation of data, the writing of the report or the decision to submit the paper for publication.

## Note

[1]　The 'Employee Insurance Agency' (Uitvoeringsinstituut Werknemers Verzekeringen) is a Dutch government body responsible for implementing employee benefit schemes.

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
