# Peer review of "A Closer Look at the Quest for an Inclusive Research Project: ‘I Had No Experience with Scientific Research, and then the Ball of Cooperation Started Rolling’"

_socsci, doi:10.3390/socsci11050186_

Round 1
Reviewer 1 Report
I am pleased to see a research team seek to study and unpack their inclusive research experience. As an emerging field, it is important to do this work so we can learn from one another and better build together conceptual and practice tools that can fuel success.
I write this review as an academic researcher in the United States. I situate myself to be transparent that my remarks are informed by this context; some of my remarks may emerge because of my greater familiarity with scholarship here; some of my remarks may emerge because of a need to better address an international audience in the writing of this manuscript.
I would like to see the authors develop greater attention to what is novel in this manuscript, with more attention given to the international scholarship related to inclusive research. This would include revisions primarily to the introduction and the discussion to better situate the current state of the science and what this work contributes to the state of the science (both in terms of leading this work's focus or purpose and in interpreting findings). Examples of scholarship that should be built upon include: Inclusive Research Network-Ireland; Autistica-UK; AASPIRE-USA; Yell Lab-USA.
There are several points that need clarification. These include:
- The introduction indicates that Author 2 has an intellectual disability whereas the methods indicate the author is autistic. While both typically classified under the broader umbrella of developmental disability, they are distinct. Since it appears the research itself focused on people with intellectual disability (who may or may not have a co-occurring label of autism), the authors should clearly address the implications that their inclusive research approach did not in fact involve a member of the community or population of interest.
While the focal research study is described by the authors as inclusive research, it appears not be universally true throughout all phases of the research project. Even in the discussion this is unclear: the authors note Bigby's classification system but then fail to us a term for their own project that is clearly anchored on this schema. The authors should clearly revise the manuscript to convey the level and nature of collaboration across all phases of the research project from conceptualization to articulation of methods to interpretation of findings and their dissemination (what roles were played and by who? how were decisions made? what elements were inclusive and what elements were not? what role did different time committments to the project play? etc.). In addressing this, the authors should address power and power dynamics, and their impact on the project and on the conclusions. This promotes transparency and more appropriate interpretation of findings given the specific research approach of the focal study.
By adding specificity and better situating this research in the context of existing scholarship, I encourage the authors to wrote more concretely so that others can learn. What did each other perceive when they felt they were compatible or a good match? For example, what shared values or traits might have enabled the establishment and maintenance of a collaborative relationship? What principles would the authors extract based on this experience?
Another related dimension is how much does extant scholarship address the professionalization of co-researchers, particularly as a viable, stable career path? Does funding context matter to this (for example, sustained funding vs. funding for specific projects)? What have others learned about the benefits (and drawbacks) of having supports external to the project such as a job coach?
- Clarify in the methods section whether coding was inductive, deductive, or a combination thereof.
- Given the increasing need for and, at times, benefits of remote collaborations it would be good to learn more specifically how the first two authors succeeded, or not, in working collaboratively and what can be gleaned for others.
- In the conclusion, the authors not not all people with disabilities are motivated ready and able to be researchers for long periods of time. Since the same is true for people without disabilities, the authors should revise this statement.
- In the discussion it is surprising to learn Author 1 one did not find personal benefits. As a PhD students and new to inclusive research, I would think Author 1 would have derived substantial benefit (for example, learning about leading research projects, learning about inclusive research, learning about the lived experience, learning about the ways working with a co-researcher impacts research, learning about community values, priorities, needs, etc.). As is, this statement suggests ableism and a lack of reflection. I encourage the authors to dig into this claim.
Author Response
Dear colleague,
Thank you for giving our manuscript “A closer look at the quest for an inclusive research project. 'I had no experience with scientific research, and then the ball of cooperation started rolling'” such thorough and constructive feedback. We did our best to address your points of feedback, within the short time frame that was available for completing the revision. We feel confident that the changes we made have resulted in a substantially improved manuscript, and we are very grateful for this opportunity.
Below you find an overview of the made changes that we made, presented along the points of feedback. In the manuscript, all changes (also those made in reaction to feedback of the two other reviewers) are typed in red. We hope this will make it easier to locate them.
Best regards,
The authors
Points of feedback and changes were made:
- I would like to see the authors develop greater attention to what is novel in this manuscript, with more attention given to the international scholarship related to inclusive research. This would include revisions primarily to the introduction and the discussion to better situate the current state of the science and what this work contributes to the state of the science (both in terms of leading this work's focus or purpose and in interpreting findings). Examples of scholarship that should be built upon include: Inclusive Research Network-Ireland; Autistica-UK; AASPIRE-USA; Yell Lab-USA.
We have revised the introduction and discussion section in order to better situate our work within the available body of research on the topic of inclusive research. For example, in the introduction we included examples of different contexts in which inclusive research practices are used (Introduction, 4th paragraph) and aligned our work with the tradition of doctoral students opting for a more inclusive approach in their research work (Introduction 6th paragraph). In the discussion, we have embedded our findings more with the work of colleagues.
- The introduction indicates that Author 2 has an intellectual disability whereas the methods indicate the author is autistic. While both typically classified under the broader umbrella of developmental disability, they are distinct. Since it appears the research itself focused on people with intellectual disability (who may or may not have a co-occurring label of autism), the authors should clearly address the implications that their inclusive research approach did not in fact involve a member of the community or population of interest.
Author 2 has a mild intellectual disability (conform described in Introduction, 6th paragraph, 2nd line), and that he has also been diagnosed with a autism spectrum disorder. In his quote in the Context section, he only mentions the latter, which may raise questions. We have solved this by adding a sentence about both being the case (2.1., 1st paragraph, line 5, Page 6).
- While the focal research study is described by the authors as inclusive research, it appears not be universally true throughout all phases of the research project. Even in the discussion this is unclear: the authors note Bigby's classification system but then fail to use a term for their own project that is clearly anchored on this schema. The authors should clearly revise the manuscript to convey the level and nature of collaboration across all phases of the research project from conceptualization to articulation of methods to interpretation of findings and their dissemination (what roles were played and by who? how were decisions made? what elements were inclusive and what elements were not? what role did different time commitments to the project play? etc.). In addressing this, the authors should address power and power dynamics, and their impact on the project and on the conclusions. This promotes transparency and more appropriate interpretation of findings given the specific research approach of the focal study.
In the discussion we have added a statement in which we position our project, at least since the collaboration with Author 2 started, as being aligned with what Bigby et al. (2014) classify as a collaborative group approach (Discussion, 5th paragraph, Page 25). Due to limited available time for revising the manuscript, we did not include an overview of the level and nature of collaboration across all phases of the research project. However, we did address this issue –as suggested- in the discussion with regard to (changes in) power relations (Discussion, 5th paragraph Pages 24-25).
- By adding specificity and better situating this research in the context of existing scholarship, I encourage the authors to wrote more concretely so that others can learn. What did each other perceive when they felt they were compatible or a good match? For example, what shared values or traits might have enabled the establishment and maintenance of a collaborative relationship? What principles would the authors extract based on this experience?
We have tried place more focus in our discussion on what our findings can mean for other researchers who wish to engage in inclusive research. See, for example, the 2nd paragraph of the discussion in which we put forward the importance of recruiting researchers and co-researchers who are motivated, ready and able to participate in an inclusive research team, and of providing education and training into inclusive research methodologies. In this paragraph we also pay attention to our positive experiences with providing support on working (as a co-researcher) through a job-coach.
- Another related dimension is how much does extant scholarship address the professionalization of co-researchers, particularly as a viable, stable career path? Does funding context matter to this (for example, sustained funding vs. funding for specific projects)? What have others learned about the benefits (and drawbacks) of having supports external to the project such as a job coach?
This is indeed an important issue, which we also address in the second paragraph in the discussion (Pages 22-23).
- Clarify in the methods section whether coding was inductive, deductive, or a combination thereof.
A comment from another reviewer made us re-think carefully about our methods: what we did and how we did this. This made us come to the conclusion that what we did was more a reflective process than a thematic content analysis (however, there were some elements of a thematic content analysis in our the reflection process). We changed the description of the Materials and methods section so that is more aligned with the followed procedures. As a result of this revision, the findings are not so much themes and sub-themes but a presentation of the reflections on our experiences with doing research together, organised along three overarching themes: preparing for the research collaboration, collaborating as a complex process and conducting research together. (Materials and Methods, 4th paragraph, Pages 9-10)
- Given the increasing need for and, at times, benefits of remote collaborations it would be good to learn more specifically how the first two authors succeeded, or not, in working collaboratively and what can be gleaned for others.
We provide more information on this in 4.2.5. (Pages 19-20)
- In the conclusion, the authors not not all people with disabilities are motivated ready and able to be researchers for long periods of time. Since the same is true for people without disabilities, the authors should revise this statement.
This is absolutely true, thank you so much for pointing this out. We have changed this sentence (Discussion, 2nd paragraph, line 4, Page 22).
- In the discussion it is surprising to learn Author 1 one did not find personal benefits. As a PhD students and new to inclusive research, I would think Author 1 would have derived substantial benefit (for example, learning about leading research projects, learning about inclusive research, learning about the lived experience, learning about the ways working with a co-researcher impacts research, learning about community values, priorities, needs, etc.). As is, this statement suggests ableism and a lack of reflection. I encourage the authors to dig into this claim.
In retrospect, we indeed feel that the benefits for the academic researcher (Author 1) have been underexposed in the interviews. During the interviews more attention was paid (unjustifiably so) to exploring the benefits for the co-researcher. This is not to say that the academic researcher did not find personal benefits. In the discussion we highlight specifically the benefits of the enhanced quality and relevance of her PhD project (3rd paragraph, final sentence, Pages 23-24).
Reviewer 2 Report
Thank you for the opportunity to read this thoughtful reflection about the inclusive research process. This manuscript adds to the growing voice of co-researchers in academic literature, as called for by Strnadova & Walmsley (2018). The manuscript is well written. I do have some concerns about the manuscript's engagement with previous/more current research and would like to see some more details added prior to publication. Overall, the manuscript makes an important contribution by bringing forward the voice of the co-researcher.
Below I provide comments section-by-section.
Abstract:
-The language "fascinating process" may be seen as otherizing.
-What is meant by "double" interview? Perhaps saying "joint" or "simultaneous" interview may be a bit clearer.
Introduction
-Around lines 43-45 on p. 1, I would like to see some additional citations supporting the claims about benefits. Perhaps each of the benefits (to co-researchers, academic researchers, and people with disabilities more broadly) could be discussed one sentence at a time?
-It would be helpful to provide a few more examples of how involvement of co-researchers can contribute to research quality. However, the editor may feel that this is not necessary for the special issue's audience.
-Line 69--what type of generalizable conclusions have other authors drawn?
Context
-It seems important to at some point explain why the previous partnership did not work out. What lessons were learned any carried forward to this project? Or, how did the previous partnership's shortcoming shape the researchers' approach to this partnership?
Materials/Methods
-Lines 148-151: Please clarify if "conversations" and "interviews" are the same here or if you are referring to two distinct processes.
-Did the co-researcher have any supports to complete the logbooks? Who provided this support? The academic researcher or the job coach?
-Please provide more information about the interview questions--please provide some examples so the reader can better understand the data collection process and how the interview questions were aligned with the study's objectives.
-What supports were provided to the co-researcher during logbook review to support recall prior to/during the interview?
-What supports were provided for the co-researcher to read and highlight the lengthy transcripts during analysis? Was the individual able to complete this task with only audio-support? Or were other supports provided?
-It would be helpful to describe some of the co-researcher's skills. It seems that this co-researcher had a very high level of literacy and verbal expression. This should be made explicit so the reader can understand how this process may not work with all co-researchers.
-To the above point, please note if there was extensive editing to the co-researcher's quotes. It seems that either this individual has a very high level of verbal expression, or the quotes were highly edited.
Findings
-Line 191--Please clarify "this" decision.
-It seems that rather than "themes," the authors organized their reflections by phases of research (e.g., preparing, conducting research, analyzing, interviewing) and specific processes (e.g., the search, role allocation). Typically, phases of research themselves wouldn't be considered themes. However, it is possible that themes emerged within each phase (e.g., "time"). I wouldn't consider processes such as "the search for a good duo" to be a theme. What recurrent ideas or concepts emerged as you more deeply reflected on this search? These recurrent ideas or concepts would be "themes."
-In section 4.1.2 please clarify what would be "suitable" characteristics for a co-researcher. What traits was the academic looking for? Why were they looking for these traits? Did literature at all guide your process of looking for a co-researcher?
-In section 4.2.2 please discuss any processes used by the pair to communicate expectations and needs. What accommodations or extra resources were involved?
-In section 4.2.2 please discuss any challenges in getting to know each other and form a working relationship. Were there any approaches that the team tried, but had to be revised?
-I appreciate you naming the tension of the academic researcher having much more time to work than the co-researcher. This is a very important point and source of power differences in the process. I would love to see this point brought forward more in the discussion.
-I would like to see reports of any conflicts experienced by the team (if any) and for these conflicts to be brought into the discussion and integrated with current literature.
Discussion
-Overall, I would like to see deeper conversation with published literature. Each of the team's findings have been discussed, in some way or another, by multiple teams. I have provided some suggested resources below, but the authors may consider others.
-The benefits of inclusive research have been discussed by many teams. I would like to see the benefits reported in this study put in better conversation with this literature.
-I also appreciate the discussion of the important role of the job coach. This seems unique. Perhaps this could be put in conversation with previous literature discussing the significant time and resources that researchers put into providing job coaching/logistical services.
-Did the team discuss power and tensions experienced? Could this be discussed?
-A limitations/future research section is needed. The authors should consider commenting on the limitation of being present for the interviews with each other and/or the knowledge that the research partners would see everything said. What may have been lost/unspoken because of this? What future research could be undertaken? In addition, what additional research questions did this study evoke for the pair?
Potential resources to consider:
Armstrong, A., Cansdale, M., Collis, A. R., Collis, B. E., Rice, S., & Walmsley, J. (2019). What makes a good self-advocacy project? The added value of co-production. Disability & Society, 34(7-8), 1289-1311.
Milner, P., & Frawley, P. (2019). From ‘on’to ‘with’to ‘by:’People with a learning disability creating a space for the third wave of inclusive research. Qualitative Research, 19(4), 382-398.
Salmon, N., Barry, A., & Hutchins, E. (2018). Inclusive research: an irish perspective. British Journal of Learning Disabilities, 46(4), 268-277.
Schwartz, A. E., Kramer, J. M., Cohn, E. S., & McDonald, K. E. (2020). “That felt like real engagement”: Fostering and maintaining inclusive research collaborations with individuals with intellectual disability. Qualitative health research, 30(2), 236-249.
Schwartz, A. E., & Durkin, B. (2020). “Team is everything”: Reflections on trust, logistics and methodological choices in collaborative interviewing. British Journal of Learning Disabilities, 48(2), 115-123.
Strnadová, I., & Walmsley, J. (2018). Peer‐reviewed articles on inclusive research: Do co‐researchers with intellectual disabilities have a voice?. Journal of Applied Research in Intellectual Disabilities, 31(1), 132-141.
Stack, E. E., & McDonald, K. (2018). We are “both in charge, the academics and self‐advocates”: Empowerment in community‐based participatory research. Journal of Policy and Practice in Intellectual Disabilities, 15(1), 80-89.
Author Response
Dear colleague,
Thank you for giving our manuscript “A closer look at the quest for an inclusive research project. 'I had no experience with scientific research, and then the ball of cooperation started rolling'” such thorough and constructive feedback. As we only had a short time frame available for completing the revision, we had to make choices regarding which feedback we would address more thoroughly in our manuscript than others. We feel confident that the changes we made have resulted in a substantially improved manuscript, and we are very grateful for this opportunity.
Below you find an overview of the made changes that we made, presented along the points of feedback. In the manuscript, all changes (also those made in reaction to feedback of the two other reviewers) are typed in red. We hope this will make it easier to locate them.
Best regards,
The authors
Changes in the manuscript:
Abstract:
- The language "fascinating process" may be seen as otherizing.
Thank you for pointing this out. We have deleted the word ‘fascinating’. (Page 1, Abstract, 4th sentence)
- What is meant by "double" interview? Perhaps saying "joint" or "simultaneous" interview may be a bit clearer.
With double interview we indeed meant to express that three out of five meetings (we changed this as this is a better word for what they were) were conducted jointly (author 1 and author 2 both participated), while in the other two meetings the moderator talked to either author 1 or author 2 (see also under ‘Materials and Method’). We followed your suggestion and substituted the word ‘double’ by ‘joint’. (Page 1, Abstract, 5th sentence)
Introduction
- Around lines 43-45 on p. 1, I would like to see some additional citations supporting the claims about benefits. Perhaps each of the benefits (to co-researchers, academic researchers, and people with disabilities more broadly) could be discussed one sentence at a time?
+
- It would be helpful to provide a few more examples of how involvement of co-researchers can contribute to research quality. However, the editor may feel that this is not necessary for the special issue's audience.
+
- Line 69--what type of generalizable conclusions have other authors drawn?
We have revised the introduction section with an aim of situating our work better within the available literature on the topic of inclusive research. For example, we have included paragraphs on examples of different contexts in which inclusive research practices are used (Introduction, 4th paragraph) and on the alignment of our work with a tradition of doctoral students opting for a more inclusive approach in their research work (Introduction, 6th paragraph). We have tried to also address the three above points of feedback. However, given the very limited timeframe we had available for this revision, we had to make a selection regarding what we elaborated on more and on what we elaborated less. We feel that the changes have improved the introduction section and we hope that the reviewer agrees.
Context
- It seems important to at some point explain why the previous partnership did not work out. What lessons were learned any carried forward to this project? Or, how did the previous partnership's shortcoming shape the researchers' approach to this partnership?
This was left out, because of privacy reasons. However, as we agree that the previous collaboration did have an impact on the collaboration between Author 1 and 2, we attempted to clarify this. We did this in the findings, 4.1.2. (1st paragraph, pages 11-12).
Materials/Methods
- Lines 148-151: Please clarify if "conversations" and "interviews" are the same here or if you are referring to two distinct processes.
Our apologies for this mix-up, we made a mistake by using the word ‘interviews’ in the last sentence of the first paragraph while this concerned conversations. We rectified this. (Materials and Methods, 2nd paragraph, last sentence, page 8)
- Please provide more information about the interview questions--please provide some examples so the reader can better understand the data collection process and how the interview questions were aligned with the study's objectives.
More information is provided on the topics and some examples of questions are provided. (Materials and Methods, 3rd paragraph, lines 3-5, page 9).
- Did the co-researcher have any supports to complete the logbooks? Who provided this support? The academic researcher or the job coach?
+
- What supports were provided to the co-researcher during logbook review to support recall prior to/during the interview?
+
- What supports were provided for the co-researcher to read and highlight the lengthy transcripts during analysis? Was the individual able to complete this task with only audio-support? Or were other supports provided?
With regard to feedback points 9-11, we added information on the provided supports regarding the completion and review of the logbooks. (3. Materals and methods, paragraphs 2, 3, and 4, Pages 8-10).
- It would be helpful to describe some of the co-researcher's skills. It seems that this co-researcher had a very high level of literacy and verbal expression. This should be made explicit so the reader can understand how this process may not work with all co-researchers.
+
- To the above point, please note if there was extensive editing to the co-researcher's quotes. It seems that either this individual has a very high level of verbal expression, or the quotes were highly edited.
During our revision process, we struggled a bit with point 12, although we certainly understand where this question comes from. Author 2 indeed has a very high level of verbal expression, and the quotes have not been extensively edited. However, we feel that pointing this out explicitly in the article places Author 2 in an unattractive/unequal position with respect to Author 1 (whose skills are not further described). That is, his level of verbal expression is clearly demonstrated by his quotes. In consultation with Author 2 we decided not to address this in the manuscript.
Findings
- Line 191--Please clarify "this" decision.
This sentence has been revised to clarify this. (4.1.1., 1st sentence, Page 10)
- It seems that rather than "themes," the authors organized their reflections by phases of research (e.g., preparing, conducting research, analyzing, interviewing) and specific processes (e.g., the search, role allocation). Typically, phases of research themselves wouldn't be considered themes. However, it is possible that themes emerged within each phase (e.g., "time"). I wouldn't consider processes such as "the search for a good duo" to be a theme. What recurrent ideas or concepts emerged as you more deeply reflected on this search? These recurrent ideas or concepts would be "themes."
We wish to thank the reviewer for this remark, as this made us re-think carefully about our methods: what we did and how we did this. This made us come to the conclusion that what we did was more a reflective process than a thematic content analysis (however, there were some elements of a thematic content analysis in our the reflection process). We changed the description of the Materials and methods section so that is more aligned with the followed procedures. As a result of this revision, the findings are not so much themes and sub-themes but a presentation of the reflections on our experiences with doing research together, organised along three overarching themes: preparing for the research collaboration, collaborating as a complex process and conducting research together. (Materials and Methods, 4th paragraph, Pages 9-10)
- In section 4.1.2 please clarify what would be "suitable" characteristics for a co-researcher. What traits was the academic looking for? Why were they looking for these traits? Did literature at all guide your process of looking for a co-researcher?
This information was added in 4.1.2, 2nd paragraph, page 132
- In section 4.2.2 please discuss any processes used by the pair to communicate expectations and needs. What accommodations or extra resources were involved?
The researchers met up with each other every week. During this meetings, they took out time to discuss whatever they were busy with and what was happening in their lives that they wanted to share (both in work and their private lives). These were informal meetings. These meetings were also the time to discuss expectations and needs, and they were frequently used for this, especially after they got to know each other well. We included this in 4.2.3., as we felt this was the best place to locate this. (4.2.3., 2nd paragraph, Page 17)
- In section 4.2.2 please discuss any challenges in getting to know each other and form a working relationship. Were there any approaches that the team tried, but had to be revised?
We focused on the factor time, as in our experience this was the most important challenge we encountered. At times we felt that it was difficult to take out enough time to connect with each other outside of research work, while was in our experience very much needed to build up and maintain a good working relationship. (4.2.2., 2nd paragraph, last sentence, Pages 15-16).
- I appreciate you naming the tension of the academic researcher having much more time to work than the co-researcher. This is a very important point and source of power differences in the process. I would love to see this point brought forward more in the discussion.
We included this issue in our discussion on power balance in inclusive research. (Discussion, 5th paragraph, line 6, Page 24)
- I would like to see reports of any conflicts experienced by the team (if any) and for these conflicts to be brought into the discussion and integrated with current literature.
We did not experience any conflicts during our collaboration. Of course, at times we disagreed on things, and there were also instances when one of us was somewhat frustrated about the collaboration. However, this was generally quickly solved by taking out the time and discussing these things openly with other. The fact that both authors felt the room to do this was probably due to the good working relationship. We did not include this in our revised manuscript.
Discussion
- Overall, I would like to see deeper conversation with published literature. Each of the team's findings have been discussed, in some way or another, by multiple teams. I have provided some suggested resources below, but the authors may consider others.
Thank you for these useful suggestions. We have revised the discussion, and embedded our findings more with the work of colleagues.
- The benefits of inclusive research have been discussed by many teams. I would like to see the benefits reported in this study put in better conversation with this literature.
We have tried to do this some more by including a few references in our discussion on benefits. (Discussion, 3rd and 4th paragraph, Pages 23-24)
- I also appreciate the discussion of the important role of the job coach. This seems unique. Perhaps this could be put in conversation with previous literature discussing the significant time and resources that researchers put into providing job coaching/logistical services.
We gave the topic of the job coach a more prominent role in our discussion section, both with respect to how to organize inclusive research (Discussion, 2nd paragraph, last 4 sentences, Page 23) and in our discussion on power relations (Discussion, 5th paragraph, 5th sentence, Page 24).
- Did the team discuss power and tensions experienced? Could this be discussed?
No, this was not discussed explicitly as such during the meetings. In the article we did discuss this topic more thoroughly. (Discussion, 5th paragraph, Page 24-25)
- A limitations/future research section is needed. The authors should consider commenting on the limitation of being present for the interviews with each other and/or the knowledge that the research partners would see everything said. What may have been lost/unspoken because of this? What future research could be undertaken? In addition, what additional research questions did this study evoke for the pair?
An extra paragraph on this was added to the discussion. (Discussion, 7th (last) paragraph, Page 26)
Reviewer 3 Report
Thank you for referring this paper to me for review. I enjoyed reading it very much. This paper provides a reflection on the inclusive research process from the perspective of the researchers involved. This in itself is not novel - there is a huge amount of research that does that already. However the depth of analysis and the description of the reflective methodology was original. There are a few things that I would suggest to improve the paper prior to publication.
The introduction and background needs to be much enhanced in its review of existing literature, rather than just describing it very briefly. Given that there is already so much literature in this area - it is important to provide a more thorough overview for readers and show how this paper adds. An overview is important because it stops each project reinventing the wheel and writing about it. Also, the paper should reference those specific papers which have already engaged in this type of reflective co-research with people with intellectual disability e.g. Bjornsdottir and Svensdottir (2008)
What is meant by the phrase at line 69, p.2 "other publications attempt to arrive at assertions that can be generalised" - generalised in what way? Frankena is cited here - the authors might like also to go back to the results Frankena's review of inclusive research in intellectual disability as part of an enhanced background literature (though I wouldn't leave it at only that).
It would have been good to engage with some of this literature with respect to questions about how to conduct inclusive research of this nature in relation to PhD projects and overall ownership of results e.g. Bjornsdottir and Svensdottir, 2008; Dorozenko et al 2016; Morgan et al 2014. How did the author manage the fact that a PhD project must be an original contribution of the student clearly delineated from others on the project? Did that impact the relationship - an important point given the concerns in the existing literature.
What is 'Philadelphia' - it was referred to throughout with no explanation.
Why did previous researchers leave? This is alluded to at several points in the article but no explanation given. This may be for privacy reasons, but as it acts as a counterpoint to the 'successful' relationship examined here it would be important to clarify if possible.
The methods section should refer to the methodology of auto-ethnography or related methods where people are interviewing or examining themselves in research. How does the approach taken here link to that? Page 10 - there is some excess text in here which I think is just the instructions to authors.
Author Response
Dear colleague,
Thank you for giving our manuscript “A closer look at the quest for an inclusive research project. 'I had no experience with scientific research, and then the ball of cooperation started rolling'” such thorough and constructive feedback. We did our best to address your points of feedback, within the short time frame that was available for completing the revision. We feel confident that the changes we made have resulted in a substantially improved manuscript, and we are very grateful for this opportunity.
Below you find an overview of the made changes that we made, presented along the points of feedback. In the manuscript, all changes (also those made in reaction to feedback of the two other reviewers) are typed in red. We hope this will make it easier to locate them.
Best regards,
The authors
Points of feedback and changes were made:
Introduction and background
- The introduction and background needs to be much enhanced in its review of existing literature, rather than just describing it very briefly. Given that there is already so much literature in this area - it is important to provide a more thorough overview for readers and show how this paper adds. An overview is important because it stops each project reinventing the wheel and writing about it. Also, the paper should reference those specific papers which have already engaged in this type of reflective co-research with people with intellectual disability e.g. Bjornsdottir and Svensdottir (2008)
We have revised the introduction (as well as the discussion) with an aim of better situating our work within the available literature on the topic of inclusive research. More specifically, in the introduction we have included paragraphs on examples of different contexts in which inclusive research practices are used (Introduction, 4th paragraph) and on the alignment of our work with a tradition of doctoral students opting for a more inclusive approach in their research work (Introduction 6th paragraph, see also feedback point 3). In the discussion, we have embedded the discussion of our findings more with the work of colleagues.
- What is meant by the phrase at line 69, p.2 "other publications attempt to arrive at assertions that can be generalised" - generalised in what way? Frankena is cited here - the authors might like also to go back to the results Frankena's review of inclusive research in intellectual disability as part of an enhanced background literature (though I wouldn't leave it at only that).
We have clarified this by adding that this concerned generalisation across inclusive research initiatives. We also added two examples with references. (Introduction, 5th paragraph, last sentence, page 4)
- It would have been good to engage with some of this literature with respect to questions about how to conduct inclusive research of this nature in relation to PhD projects and overall ownership of results e.g. Bjornsdottir and Svensdottir, 2008; Dorozenko et al 2016; Morgan et al 2014. How did the author manage the fact that a PhD project must be an original contribution of the student clearly delineated from others on the project? Did that impact the relationship - an important point given the concerns in the existing literature.
This is an excellent suggestion, thank you for this. We have added an extra paragraph on this topic (Introduction, 6th paragraph, pages 4-5).
- What is 'Philadelphia' - it was referred to throughout with no explanation.
Philadelphia is the name of the service organisation where the authors work. This name should have been blinded for review, but by mistake it was still there in the quote of author 2. In the same paragraph this name is also mentioned and explained (but there it was blinded), so in the un-blinded manuscript there will not be any confusion on this. Our sincere apologies for this mistake. (2.1., 1st paragraph, Page 6).
- Why did previous researchers leave? This is alluded to at several points in the article but no explanation given. This may be for privacy reasons, but as it acts as a counterpoint to the 'successful' relationship examined here it would be important to clarify if possible.
Yes indeed, we left this out due to privacy reasons. However, as we agree that these previous collaborations did have an impact on the final collaboration between Author 1 and 2, we attempted to clarify this some more. We did this in two places: section 2.2 (2nd paragraph, page 7) and, most importantly, in 4.1.2. of the findings (1st paragraph, pages 11-12).
Methods
- The methods section should refer to the methodology of auto-ethnography or related methods where people are interviewing or examining themselves in research. How does the approach taken here link to that?
We have added this to the description of our methods (3. Materials and Methods, 1st paragraph, Page 8).
- Page 10 - there is some excess text in here which I think is just the instructions to authors.
We did not see any excess text in our manuscript. We hope (and trust) in that this is not the case in the revised manuscript.
Round 2
Reviewer 2 Report
Thank you to the authors for your thorough revisions. The authors have made great efforts to revise this manuscript. I still have some comments that I hope continue to help the authors situate their work within the broader discussions in our field. Your thorough description of the partnership formation and challenges are a strong contribution.
INTRODUCTION
1) Page 1-you say "mutual understanding" is a benefit. What is this a mutual understanding about? Research? Lived experiences? Something else?
2) I would like to see the authors acknowledge a greater breadth of contributions that co-researchers make beyond making appropriate materials in their example on p. 2. The literature points to many other contributions.
3) I appreciate that the authors have incorporated/referenced various research teams throughout the world. However, I believe I and the other reviewers were hoping for more than a listing of these teams. The topic sentence of this paragraph states that "inclusive research takes different forms in different contexts." However, the remaining sentences in this paragraph are not focused on how practices differ, but rather say that these teams exist. Please revise to comment on the different processes, or clearly state that this paragraph is intended to list examples of other teams doing inclusive research. Additionally, for each of these teams, there may be additional and potentially more apt citations for the points the authors are trying to make.
4) This is not necessary, but you may consider adding some more citations/reading when thinking about publications that "attempt to arrive at assertions that can be generalized." You may find that each of these publications has additional ideas that you hint at in your discussion section.
Nicolaidis, C., Raymaker, D., Kapp, S. K., Baggs, A., Ashkenazy, E., McDonald, K., Weiner, M., Maslak, J., Hunter, M., & Joyce, 674 A. (2019). The AASPIRE practice-based guidelines for the inclusion of autistic adults in research as co-researchers and study 675 participants. Autism : the international journal of research and practice 23: 2007–2019. DOI: 10.1177/1362361319830523
Schwartz, A. E., Kramer, J. M., Cohn, E. S., & McDonald, K. E. (2020). “That felt like real engagement”: Fostering and maintaining inclusive research collaborations with individuals with intellectual disability. Qualitative health research, 30(2), 236-249.
Stack, E. E., & McDonald, K. (2018). We are “both in charge, the academics and self‐advocates”: Empowerment in community‐based participatory research. Journal of Policy and Practice in Intellectual Disabilities, 15(1), 80-89.
5) Overall, nice job taking the feedback provided by all 3 reviewers in this section
CONTEXT
1) Since the author names themselves as having PDD-NOS and autism, might this be their main disability identity, rather than intellectual disability? Perhaps discussing with the co-researcher what label they would prefer may help clarify (as all three reviewers had questions). I understand that the author has both intellectual disability and autism diagnoses. However, I want to be sure that they identify with intellectual disability, since their own narrative did not mention it. If they prefer the label of autism, then I would note that and remove references to them as having an intellectual disability.
METHODS
1) This section is much improved, as the authors have provided important details about their work together.
2) Did author 2 receive any support (and from who) to review their logbook?
3) How was the data analyzed? It seems that the authors have explained how they moved from transcripts to some "main ideas." However, I'm still not sure how they arrived at their themes. What was the decision making process (with other researchers), as referenced on p. 5? Please add additional information about your analytical process (including methodological citations, if applicable).
FINDINGS
1) I appreciate the additional contextual information about the how the partnership was formed. This is very helpful.
2) Section 4.2.5: What supports were provided to author 2 during remote/virtual work? Did this individual code the transcripts independently? Or receive support? Were any tools used to replicate the use of post-it notes? Or were transcripts just coded without these aids? The challenge named seems to imply that no such tools were used. Perhaps clearly stating that coding was limited to word processing software or printed transcripts would be helpful in understanding the process.
DISCUSSION
1) Another reviewer commented that the PhD student did not clearly name benefits that she received. I see there are some additions here, but they seem mostly focused on satisfaction in having provided benefits to author 2. What did the student (author 1) learn? How did this process further her training and/or understanding of knowledge production processes? How did this shape her education?
2) "In this regard, we share the opinion of Björnsdóttir and 566 Svensdóttir (2008) that well-executed inclusive research is of the same value as non-inclusive research" -This statement seems to be justifying inclusive research, as if the authors expect readers to doubt the value of inclusive research (or that they themselves may have doubted it!). Might this be reframed? Might the authors want to talk about the STRENGTHS of their process, not just that it is "as good"?
Author Response
Thank you for your kind words about our manuscript, which indeed has already greatly improved which is also thanks to your previous comments. We also want to thank you for your new comments that will help to improve the manuscript even more. Below, we address your comments one by one. In the revised manuscript (revision 2), the new revisions that have been made in response to your comments have again been in written in red.
INTRODUCTION
1)Page 1-you say "mutual understanding" is a benefit. What is this a mutual understanding about? Research? Lived experiences? Something else?
Authors: Thank you for pointing this out, indeed this is probably not clear for readers. We changed this into: “gaining insight into the experiences of (other) people with intellectual disabilities”, because this describes the point which we intended to make better.
* Change: Page 2, lines 20-21.
2) I would like to see the authors acknowledge a greater breadth of contributions that co-researchers make beyond making appropriate materials in their example on p. 2. The literature points to many other contributions.
Authors: Again, we would like to thank the reviewer for this helpful comment. We addresses this by extending our attention from the technical-instrumental contribution of researchers with intellectual disabilities (i.e. contributing to appropriate research materials) to also include important gains for a research team (by concretizing abstract terms) ánd the great talents of persons with i.d. for pointing out barriers and good practices (using the article of Chalachanova and Gjermestad, 2021 as a reference).
*Change: Pages 2 (last 2 lines)-3 (lines 1-7).
3) I appreciate that the authors have incorporated/referenced various research teams throughout the world. However, I believe I and the other reviewers were hoping for more than a listing of these teams. The topic sentence of this paragraph states that "inclusive research takes different forms in different contexts." However, the remaining sentences in this paragraph are not focused on how practices differ, but rather say that these teams exist. Please revise to comment on the different processes, or clearly state that this paragraph is intended to list examples of other teams doing inclusive research. Additionally, for each of these teams, there may be additional and potentially more apt citations for the points the authors are trying to make.
Authors: We have addressed your valid criticism that it remained nothing more than an enumeration in origin. To replace it, we have focused on two longer-established networks. In doing so, we have emphasized the lessons that can be learned from such long-standing practices. The insights that can be carried across networks with different co-researchers are also hinted at here.
*Change: Pages 3 (last 3 lines) and 4 (lines 1-31).
4) This is not necessary, but you may consider adding some more citations/reading when thinking about publications that "attempt to arrive at assertions that can be generalized." You may find that each of these publications has additional ideas that you hint at in your discussion section.
Nicolaidis, C., Raymaker, D., Kapp, S. K., Baggs, A., Ashkenazy, E., McDonald, K., Weiner, M., Maslak, J., Hunter, M., & Joyce, 674 A. (2019). The AASPIRE practice-based guidelines for the inclusion of autistic adults in research as co-researchers and study 675 participants. Autism : the international journal of research and practice 23: 2007–2019. DOI: 10.1177/1362361319830523
Schwartz, A. E., Kramer, J. M., Cohn, E. S., & McDonald, K. E. (2020). “That felt like real engagement”: Fostering and maintaining inclusive research collaborations with individuals with intellectual disability. Qualitative health research, 30(2), 236-249.
Stack, E. E., & McDonald, K. (2018). We are “both in charge, the academics and self‐advocates”: Empowerment in community‐based participatory research. Journal of Policy and Practice in Intellectual Disabilities, 15(1), 80-89.
Authors: Thank you for this suggestion. We took up your advice and added a reference to Schwartz et al. (2020).
*Change: Page 5, lines 7-8.
5) Overall, nice job taking the feedback provided by all 3 reviewers in this section
Authors: Thank you for this compliment.
CONTEXT
Since the author names themselves as having PDD-NOS and autism, might this be their main disability identity, rather than intellectual disability? Perhaps discussing with the co-researcher what label they would prefer may help clarify (as all three reviewers had questions). I understand that the author has both intellectual disability and autism diagnoses. However, I want to be sure that they identify with intellectual disability, since their own narrative did not mention it. If they prefer the label of autism, then I would note that and remove references to them as having an intellectual disability.
Authors: Author 2 has indeed been diagnosed with both (autism and intellectual disability), but he prefers to position himself to the world/others as a person with autism. This does not mean that he wished to hide the other diagnose or that the does not identify at all with intellectual disability. We (including author 2 himself) prefer not to not take away the reference to him being diagnosed with intellectual disability especially because this is the link with him receiving services and support from the organisation that was involved in this research.
METHODS
This section is much improved, as the authors have provided important details about their work together.
1) Did author 2 receive any support (and from who) to review their logbook?
Authors: We elaborated on this by explaining our preparations for reviewing the logbooks and the activity itself in more detail.
* Change: Page 10, lines 8-13
2) How was the data analyzed? It seems that the authors have explained how they moved from transcripts to some "main ideas." However, I'm still not sure how they arrived at their themes. What was the decision making process (with other researchers), as referenced on p. 5? Please add additional information about your analytical process (including methodological citations, if applicable).
Authors: We described our process of data processing (which may be maybe a more suitable word than data analysis) in more detail. We hope that this will satisfy the need for more information.
*Change: Page 11, lines 3-9.
FINDINGS
I appreciate the additional contextual information about the how the partnership was formed. This is very helpful.
1) Section 4.2.5: What supports were provided to author 2 during remote/virtual work? Did this individual code the transcripts independently? Or receive support? Were any tools used to replicate the use of post-it notes? Or were transcripts just coded without these aids? The challenge named seems to imply that no such tools were used. Perhaps clearly stating that coding was limited to word processing software or printed transcripts would be helpful in understanding the process.
Authors: Yes, during the period when COVID-19 restrictions were in place, the two researchers coded the transcripts independently from each other, with there being further support for author 2 (the researcher with an intellectual disability). This turned out to be possible, as author 2 (the researcher with intellectual disability) had already build up extensive experience in analysing transcripts from working on several previous studies during the research project, and they both felt comfortable in doing this independently from each other. This time no post its were used to record codes. Instead, transcripts were printed, pieces of text were highlighted and codes were notes in the margins. More information on this is added to the manuscript.
*Changes: Page 21, lines 5-9
DISCUSSION
- Another reviewer commented that the PhD student did not clearly name benefits that she received. I see there are some additions here, but they seem mostly focused on satisfaction in having provided benefits to author 2. What did the student (author 1) learn? How did this process further her training and/or understanding of knowledge production processes? How did this shape her education?
Authors: We elaborated on this by explaining that the PhD student indeed also learned from their long-term collaboration regarding ways in which they could best work together and conduct research together. In this regard, it were especially the difficult situations that provided an opportunity to learn. As an example, we refer to a previous collaboration which was halted because the roles of being a supporter and being a colleague became too much intertwined.
*Change: Page 25, lines 14-18.
2) "In this regard, we share the opinion of Björnsdóttir and 566 Svensdóttir (2008) that well-executed inclusive research is of the same value as non-inclusive research" -This statement seems to be justifying inclusive research, as if the authors expect readers to doubt the value of inclusive research (or that they themselves may have doubted it!). Might this be reframed? Might the authors want to talk about the STRENGTHS of their process, not just that it is "as good"?
Authors: We would like to thank the reviewer for alerting us to the possible misinterpretations that could be linked to this sentence. It will be clear that we (some of us who have been trying to conduct inclusive research since 1999) do not doubt the added value of inclusive research. We mainly wanted to point out the quality that was added to the research through the inclusive collaboration and that it was just these extras that took this research to the highest academic level. We have amended the text to reflect this.
*Change: Page 25 (last three lines) and 26 (first line).